# Revisiting Value Estimation in Policy Gradient Methods

## Abstract

Temporal-difference (TD) estimation is a central component of value estimation in reinforcement learning. However, its role within policy gradient methods has not been systematically understood. In this work, we introduce a framework grounded in the notion of well-posedness, which provides a rigorous formulation of TD estimation across a wide range of control problems and enables more accurate value estimation. Through extensive empirical studies, we further show that when policy optimization is properly formulated and combined with an appropriate bootstrapping strategy, even the vanilla policy gradient algorithm can reliably solve the problem. These findings indicate that deep reinforcement learning methods can be made more robust and interpretable with proper problem formulation.

## 1 Introduction

Deep reinforcement learning (RL) has emerged as a core methodology in control and robotics (Li et al., 2023; Song et al., 2023; Sleiman et al., 2024). While much of the recent progress has focused on improving the policy update mechanism, such as incorporating KL constraints in TRPO or probability ratio clipping in PPO (Schulman et al., 2015a; 2017), comparatively little attention has been devoted to value estimation within policy gradient methods. The most influential framework in this direction is Generalized Advantage Estimation (GAE) (Schulman et al., 2015b), which highlights the importance of bias–variance trade-offs. However, a systematic understanding of value estimation in policy gradient methods is still lacking.

A common view is that the role of value functions in deep policy gradient methods is primarily to serve as a baseline for reducing the variance of the gradient estimator. Recent work, however, suggests that accurate value estimation indeed plays a fundamental role in the effectiveness of modern policy gradient algorithms (Wang et al., 2025). It demonstrates that simply increasing the number of value updates per iteration allows vanilla policy gradient (VPG) to achieve performance comparable to PPO on the Gymnasium benchmark. Nevertheless, it remains unclear how the value estimation module, specifically the temporal-difference (TD) method (Sutton, 1988), operates within deep policy gradient methods. Prior work has primarily focused on the Bellman equation, which can be viewed as one-step TD estimation. Its global convergence has been established under the assumption that the state space is either finite or ergodic (Maei et al., 2009; Yu & Bertsekas, 2009; Agarwal et al., 2021). In practice, however, these assumptions rarely hold in real-world control environments. Furthermore, most existing convergence results for TD estimation are derived with respect to certain $L_2$ norm. While theoretically well-founded, such results have limited practical relevance, as they typically require a sufficient sample coverage over the entire state space.

Therefore, rather than pursuing guarantees in the $L_2$ sense, we adopt a trajectory-based perspective. We begin with a counterexample demonstrating that both the Bellman equation and TD estimation admit non-unique solutions when no terminal reward is provided. The key insight is that each trajectory should be assigned a ground-truth target value at the end of the rollout, which can then be propagated backward through the temporal-difference equation to calibrate earlier value estimates. In the absence of such a terminal value, the corresponding TD estimation for the entire trajectory becomes under-determined, leading to non-unique solutions.

A systematic understanding of value estimation also has important practical implications. Although PPO has become the *de facto* deep policy gradient algorithm, the discrepancies between the original implementation (Schulman et al., 2017) and its many variants (Makoviichuk & Makoviychuk, 2021;

Huang et al., 2022; Serrano-Muñoz et al., 2023) are so substantial that they can often be regarded as distinct algorithms. For instance, while the original PPO formulation relies on ratio clipping to restrict policy updates, in practice this mechanism alone is frequently insufficient to prevent over-updates. As a result, additional techniques such as adaptive learning rates and early stopping are commonly employed. Beyond value estimation parameters such as the discount factor $\gamma$ and the GAE factor $\lambda$, PPO further depends on hyperparameters including the number of epochs and mini-batches, clipping ratio $\epsilon$, entropy coefficient, and target KL threshold, all of which influence the dynamics of policy updates (Engstrom et al., 2020; Andrychowicz et al., 2021). This lack of a deeper understanding of how these components interact contributes to the challenges of reliably implementing PPO in practice.

The success of VPG on Gymnasium suggests that, with accurate value estimates, challenging tasks can be solved by a much simpler RL algorithm, thus avoiding the extensive hyperparameter tuning required by PPO. In this work, we advance this perspective by offering a comprehensive understanding of temporal-difference estimation through the lens of well-posedness. In particular, we show that for a broad class of control problems, including locomotion and manipulation tasks, there always exists a suitable terminal reward function together with a corresponding bootstrapping method that renders TD estimation well-posed. Moreover, through extensive experiments, we demonstrate that a slightly modified VPG algorithm equipped with this well-posed value estimation scheme can effectively solve these control problems.

## 2 RELATED WORK

**Value estimation in deep RL.** The concept of value functions was originally proposed in optimal control to measure how well a policy performs at a given state, guiding policy improvement through dynamic programming (Bellman, 1957). In the context of RL, however, it is usually impossible to access to the true value function due to the curse of dimensionality, which motivates temporal difference methods that estimate the value function through Bellman equation (Sutton & Barto, 1998). It has been clearly demonstrated that accurate value estimation is critical to the performance of off-policy algorithms, particularly with the use of double Q-learning in SAC (Haarnoja et al., 2018) and TD3 (Fujimoto et al., 2018), as compared to their counterpart DDPG (Lillicrap et al., 2015). In contrast, most existing work on on-policy algorithms focuses on improving the policy objective, with the value function primarily serving to estimate the advantage and reduce the variance of $Q$ values (Peters & Schaal, 2008; Schulman et al., 2015b). Moreover, little attention has been given to understanding how policy gradient methods practically estimate values, which is one of the main problems studied in this work.

**Bootstrapping schemes.** The idea of bootstrapping at the terminal state of each rollout is well established in both optimal control and reinforcement learning. For example, in model predictive control (MPC), incorporating a terminal cost function has been shown to improve decision quality while reducing planning time (Zhong et al., 2013). Terminal costs also play a central role in many approaches that integrate MPC with RL (Bhardwaj et al., 2021; Hansen et al., 2022; 2024; Bian et al., 2025). Unlike time-dependent MPC, however, RL typically assumes a Markov framework, which poses challenges when handling timeouts. One proposed solution is to parameterize the critic as a joint function of state and time, $V = V(s, t)$ (Pardo et al., 2018). While effective in principle, this approach violates the Markov property because it requires rollouts to revisit a given state $s$ at a specific time $t$, which is often impractical. Another solution is to always bootstrap from the critic at timeout (Rudin et al., 2021), but as we later show, this formulation is ill-posed and can converge to incorrect solutions.

**Implementation of policy gradient methods.** The core advancement of policy gradient methods, particularly in modern algorithms such as TRPO (Schulman et al., 2015a) and PPO (Schulman et al., 2017), is widely attributed to the development of policy update methods inspired by conservative policy updates (Kakade & Langford, 2002). Despite their success, it has been shown that deep policy gradient algorithms often face practical challenges, including difficulties with reproducibility, sensitivity to hyperparameters, and a lack of robustness (Henderson et al., 2018). Recent work highlights that the performance of a given algorithm is heavily influenced by code-level implementation details, which are often overlooked (Engstrom et al., 2020; Andrychowicz et al., 2021).

## 3 BACKGROUND

**Policy gradient methods.** The vanilla policy gradient (VPG) algorithm maximizes the following objective

$$L^{VPG}(\theta) = \hat{\mathbb{E}}_{(s_t, a_t) \sim \pi_\theta} \Big[ \log \pi_\theta(a_t|s_t) \ \hat{A}_t \Big], \tag{1}$$

where $\hat{A}_t = \hat{Q}_t - \hat{V}(s_t)$ is the estimated advantage of $(s_t, a_t)$, and $\hat{V}$ is the estimated value function of the current policy $\pi$ which serves as the baseline. Let $V_\phi(s) = V(s; \phi)$ denote the value approximation to the true value function $V^{\pi_\theta}$ by a neural network where $\phi \in \mathbb{R}^M$ is the network parameter. Motivated by the idea that small policy updates usually guarantee policy improvement (Kakade & Langford, 2002), Schulman et al. (2015a) proposes the TRPO objective:

$$\max_\theta \quad \hat{\mathbb{E}}_{(s_t, a_t) \sim \pi_\theta} \Big[ \frac{\pi_\theta(a_t|s_t)}{\pi(a_t|s_t)} \hat{A}_t \Big], \tag{2}$$

$$\text{subject to} \quad \hat{\mathbb{E}} \Big[ D_{KL}(\pi_\theta(\cdot|s) \ \| \ \pi(\cdot|s)) \Big] \leq \delta, \tag{3}$$

where $\delta > 0$ is the tolerance. PPO (Schulman et al., 2017) simplifies the implementation by using a ratio-clipping objective:

$$L^{PPO}(\theta) = \hat{\mathbb{E}}_\pi \Big[ \min \Big( \frac{\pi_\theta(a_t|s_t)}{\pi(a_t|s_t)} \hat{A}_t, \text{clip}(\frac{\pi_\theta(a_t|s_t)}{\pi(a_t|s_t)}, 1 - \epsilon, 1 + \epsilon) \hat{A}_t) \Big) \Big], \tag{4}$$

where $\epsilon$ is the clipping parameter. Throughout this paper, we denote the current and old policy by $\pi_\theta$ and $\pi$, respectively.

**Temporal-difference (TD) estimation.** In practice, there is usually a large variance when evaluating expectation $\hat{\mathbb{E}}[\cdot]$ over a long trajectory. Sutton & Barto (1998) proposed the temporal-difference scheme that combines the idea of Monte Carlo and dynamic programming. That is, given a single trajectory $(s_0, ..., s_H)$ and the corresponding actions $(a_0, ..., a_{H-1})$, where $H$ is the horizon of TD estimation. The target value is estimated through

$$\hat{V}_t = \sum_{k=t}^{H-1} \gamma^{k-t} R(s_k, a_k) + \gamma^{H-t} V_\phi(s_H) \tag{5}$$

where $H$ is the rollout horizon and $\gamma$ is the discount factor. This estimation applies if the trajectory does not terminate within the next $H - t$ steps.

## 4 ANALYSIS OF TEMPORAL-DIFFERENCE AND BOOTSTRAPPING

It is widely accepted that the role of critic networks is to reduce the variance as a baseline. Theoretically, using $\hat{Q}_t$ and $\hat{A}_t$ provide the same gradient estimator:

$$\hat{\mathbb{E}}_{(s_t, a_t) \sim \pi_\theta} \Big[ \nabla \log \pi_\theta(a_t|s_t) \ \hat{Q}_t \Big] = \hat{\mathbb{E}}_{(s_t, a_t) \sim \pi_\theta} \Big[ \nabla \log \pi_\theta(a_t|s_t) \ \hat{A}_t \Big]$$

since $\hat{\mathbb{E}}_{a_t \sim \pi_\theta} \Big[ \nabla \log \pi_\theta(a_t|s_t) \ V_\phi(s_t) \Big] = 0$ for any $s_t$ and continuous probability density function $\pi_\theta$. However, it is difficult to fully account for the strength of the value network in practice as the previous equalities hold only when the expectations are exact, which is impossible for most control problems. To better understand, we develop a general framework to understand under which conditions the value function approximated from TD estimation accurately reflect the true value function.

**Well-posedness of TD estimation.** For policy gradient approaches, the target value is calculated along the rollout trajectory. Without loss of generality, let us consider the Bellman equation

$$V(s) = \mathbb{E}_{a \sim \pi_\theta(\cdot|s), s' \sim \mathcal{M}(s, \cdot, a)} \Big[ R(s, a) + \gamma V(s') \Big] \tag{6}$$

which can be regarded as the one-step TD estimation. Although the existence and uniqueness of the solution of Equation 6 is proved for finite MDPs (Bellman, 1957), the following counterexample shows that it is not always true for continuous MDPs:

**Example 4.1.** *Let $\mathcal{S} = [0, 1]$ and for any state $s_t \in (0, 1)$, the next state is always $s_{t+1} = \frac{s_t}{2}$ regardless of the action, and the reward model $R(s, a) \equiv 0$. The boundary values are $V(0) = V(1) = 0$. Note that for any initial state $s \in (0, 1)$, the trajectory starting from $s$ never hits $\partial\mathcal{S}$ within finite time. For any policy $\pi$, its value function is clearly $V^\pi \equiv 0$ which is also the desired solution of Bellman equation. However, one can easily verify that*

$$V(s) = \begin{cases} \gamma^{-k}, & s = 2^{-k}, k = 1, 2, ... \\ 0, & otherwise \end{cases}$$

*is also the solution of equation 6.*

In fact, it has infinitely many solutions since the trajectory starting at $s_0 = 1/2$ does not hit the boundary within finite time, thus providing no ground-truth estimation to propagate backward and calibrate earlier target values. On the other hand, if a trajectory $(s_0, ..., s_T)$ stops in $T$ steps with a terminal reward $\Phi(s_T)$ applied to the final state $s_T$, then the target value $\hat{V}(s_t) = \sum_{k=t}^{T-1} \gamma^{k-t} R(s_k, a_k) + \gamma^{T-t} \Phi(s_T)$ provides a ground-truth estimation to the real value. To further make it precise and more relevant to real algorithms, we introduce the concept of stopping time:

**Definition 4.1.** *(Stopping time) Let $\{\mathcal{N}_t\}$ be an increasing family of $\sigma$-algebras in $\Omega$ (i.e., $\mathcal{N}_r \subset \mathcal{N}_s$ if $r \leq s$). A function $\tau : \Omega \to [0, \infty]$ is called a stopping time with respect to $\{\mathcal{N}_t\}$ if*

$$\{\omega \in \Omega : \tau(\omega) \leq t\} \in \mathcal{N}_t$$

*for all $t \geq 0$.*

It can be easily verified that the truncation condition $\tau(\omega) \equiv T$ is a stopping time. Also, the first time a trajectory leaves the feasible region $\mathcal{S}$, namely $\tau_\mathcal{S} = \inf\{t \geq 0 : s_t \notin \mathcal{S}\}$, is also a stopping time which typically corresponds to the termination condition. The following theorem proves that TD estimation is guaranteed to have unique solution if all trajectories generated by $\pi$ stops within a finite time with probability 1:

**Theorem 4.1.** *(Well-posedness) Suppose that $\mathcal{S} \subset \mathbb{R}^n$ is a Borel set and $\tau$ is a stopping time with $P(\tau < \infty) = 1$. If the reward model $R$ is bounded over $\mathcal{S} \times \mathcal{A}$, then the Bellman equation 6 has unique solution*

$$V(s) = \mathbb{E}_{(s_k, a_k) \sim \pi_\theta, s_0 = s} \Big[ \sum_{k=0}^{\tau-1} \gamma^k R(s_k, a_k) + \gamma^\tau \Phi(s_\tau) \Big] \tag{7}$$

*for all $s \in \mathcal{S}$ where $\Phi(\cdot)$ is the terminal reward.*

The finite stopping time assumption $P(\tau < \infty) = 1$ is critical as it rules out the possibility of infinite rollouts shown in Example 4.1, thereby ensuring the uniqueness of solutions to both the Bellman equation and TD estimation.

**Specifying the terminal reward $\Phi(\cdot)$.** Unlike the reward function, which is explicitly defined as part of the policy optimization problem, the terminal reward function $\Phi$, despite its critical role in accurate value estimation, is typically not specified. As a result, an additionally defined $\Phi$ may introduce bias into value estimation, since the solution of Equation 7 is *not necessarily* a solution of Equation 6 with an arbitrary terminal reward $\Phi$. The following theorem establishes that there always exists a terminal reward $\Phi^*$ that recovers the ground-truth value function:

**Theorem 4.2.** *Suppose that $\mathcal{S} \subset \mathbb{R}^n$ is a Borel set and $\tau$ is a stopping time with $P(\tau < \infty) = 1$. Assume that the reward model $R$ is bounded over $\mathcal{S} \times \mathcal{A}$. Let $V^\pi$ be the value function of $\pi$, then the unique solution of Equation 7 is $V(s) = V^\pi(s)$ for all $s \in \mathcal{S}$ when $\Phi = V^\pi$.*

The proof of the above theorem follows directly from the dynamic programming principle. In practice, however, obtaining the ground-truth value function $V^\pi$ *prior to* performing TD estimation is generally infeasible. Policy gradient algorithms typically terminate a trajectory and reset the environment either when a termination condition is triggered—often corresponding to hard constraints such as collisions or unsafe states—or when the maximum episode length is reached. In the former case, the trajectory *legitimately* ends, as continuing would violate the constraints; thus, we can simply set $\Phi(s) = 0$ for any state $s$ outside the feasible domain. In contrast, termination due to timeout

| Method | Target Value | Well-posed | Time-independent |
|--------|-------------|------------|------------------|
| None | $\hat{V}(s_t) = \sum_{k=t}^{T-1} \gamma^{k-t} R(s_k, a_k)$ | *Yes* | *No* |
| Critic | $\hat{V}(s_t) = \sum_{k=t}^{T-1} \gamma^{k-t} R(s_k, a_k) + \gamma^{T-t} V_\phi(s_T)$ | *No* | *Yes* |
| Reward | $\hat{V}(s_t) = \sum_{k=t}^{T-1} \gamma^{k-t} R(s_k, a_k) + \frac{\gamma^{T-t}}{1-\gamma} R_s(s_T)$ | *Yes* | *Yes* |

Table 1: Bootstrapping methods at the terminal state.

is not dictated by the problem itself but by practical considerations such as sample efficiency and limiting excessively long trajectories. A trajectory reaching the time limit does not necessarily constitute a natural stopping point, and therefore, the treatment of TD estimation at the final step is crucial for overall value approximation accuracy.

There are two common bootstrapping methods used at timeout. The first is to do nothing at the terminal state (Brockman et al., 2016): $\hat{V}(s_t) = \sum_{k=t}^{T-1} \gamma^{k-t} R(s_k, a_k)$ which is equivalent to setting $\Phi \equiv 0$ and is therefore well-posed. However, the target value in this case depends on its position within the trajectory and is time-dependent. Another limitation arises when the reward function takes negative values: short trajectories may achieve higher returns early in training, potentially misleading the agent toward sub-optimal solutions. The second method is to bootstrap through the critic network (Rudin et al., 2021): $\hat{V}(s_t) = \sum_{k=t}^{T-1} \gamma^{k-t} R(s_k, a_k) + \gamma^{T-t} V_\phi(s_T)$ which corresponds to the original Bellman equation 6 without a terminal reward. While this formulation is time-independent, it is ill-posed: the rollout stops at $s_{T+1}$ and the environment is reset, meaning there is *no* trajectory starting from $s_{T+1}$ that provides ground-truth values to calibrate $V_\phi(s_{T+1})$. Consequently, the incorrect target value propagates backward through TD estimation, corrupting value predictions for the entire trajectory.

Motivated by the terminal cost formulation in model predictive control (MPC), we propose a new method that bootstraps through the reward function:

$$\hat{V}(s_t) = \sum_{k=t}^{T-1} \gamma^{k-t} R(s_k, a_k) + \frac{\gamma^{T-t}}{1-\gamma} R_s(s_T) \tag{8}$$

with the terminal reward $\Phi(s) = (1-\gamma)^{-1} R_s(s)$ where we assume that the reward function can be decomposed into the state and action reward, i.e., $R(s,a) = R_s(s) + R_a(a)$, and use the re-scaled state reward as the terminal reward. It is well-posed and time-independent, but relies on the assumption that the reward landscape approximately reflects the ground-truth value function $V^\pi$, as is the case in the manipulation tasks considered later. All three schemes are summarized in Table 1.

**Under-determined system.** Here we briefly illustrate the practical issues that can arise when the bootstrapping scheme is not well-posed. Consider, for example, a trajectory of length $T$, $(s_0, \ldots, s_T)$, where the value network is trained to minimize the following mean-squared error

$$\mathcal{L}(\phi) = \sum_{t=0}^{T-1} |V_\phi(s_t) - [\sum_{k=t}^{T-1} R(s_k, a_k) + \gamma^{T-t} V_\phi(s_T)]|^2 \tag{9}$$

using the Critic bootstrapping method. This objective, however, defines an under-determined system: it has $T + 1$ unknowns, $(V_\phi(s_0), \ldots, V_\phi(s_T))$, but only $T$ equations. Consequently, there exist infinitely many value predictions that achieve zero loss in Equation 9, and minimizing this loss provides no guarantee of recovering the ground-truth value function. As a result, the convergence of $\mathcal{L}(\phi)$ is not robust and can be highly sensitive to factors such as network initialization and optimization details. In contrast, for well-posed methods, the bootstrapping term $V_\phi(s_T)$ is replaced by the known terminal reward $\Phi(s_T)$, reducing the number of unknowns to match the number of equations and thereby ensuring a unique solution. This guarantees that, under any reasonably effective optimization approach and initialization, the value network should converge to this unique solution.

## 5 HOW BOOTSTRAPPING SCHEMES AFFECT PERFORMANCE?

We have theoretically analyzed the well-posedness of TD estimation and the three bootstrapping schemes, we now examine their practical implications. Specifically, we use two representative problems, PickCube and Hopper, as motivating examples to illustrate how the bootstrapping methods affect algorithm performance.

**Short-horizon manipulation tasks.** Consider the PickCube task in ManiSkill3, in which the objective is for a robot to grasp a cube and move it to a target goal position within 50 environment steps. The reward function, $R(s) = R_1(s) + R_2(s) + R_3(s)$, consists of three components, each of which explicitly characterizes the distance to the goal state at a given stage: $R_1 = 1 - \tanh(5x)$, where $x$ is the distance between the gripper and the cube; $R_2 = (\text{if grasped}) \cdot (1 - \tanh(5y))$, where $y$ is the distance between the cube and the goal position; and $R_3 = (\text{if reached}) \cdot (1 - \tanh(5z))$, where $z$ is the velocity of the gripper. We compare three bootstrapping methods using the PPO implementation from Tao et al. (2025). As shown in Figure 1 (a)-(b), the None method fails entirely, while the Critic method achieves a success rate of only around $0.2$ due to its ill-posedness, which leads to incorrect solutions. In contrast, the Reward method successfully completes the task, as the reward function $R(s)$ serves as a heuristic that reflects the distance between the current and goal states, and therefore shares the same landscape as the ground-truth value function $V^\pi(s)$.

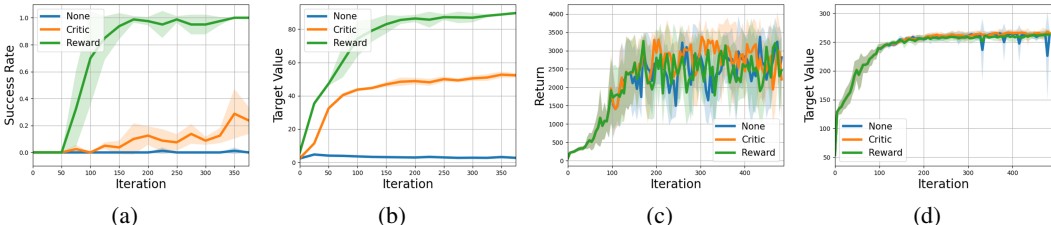

(a)           (b)           (c)           (d)

Figure 1: (a) Success rate of PickCube; (b) Estimated target value of PickCube; (c) Cumulative return of Hopper; (d) Estimated target value of Hopper. For each task, all bootstrapping methods are repeated on fixed 5 random seeds. We use the discount factor $\gamma = 0.99$ and GAE factor $\lambda = 0.95$ in all experiments, in contrast to the default setting $\gamma = 0.8$ and $\lambda = 0.9$ in Maniskill3.

**Locomotion with simple rewards.** A key distinction between manipulation and locomotion tasks is that manipulation tasks typically have a fixed trajectory length and no termination conditions, whereas locomotion tasks involve long horizons and hard constraints to limit unnecessary exploration. As a result, the overall return of a trajectory in locomotion tasks is strongly correlated with its length. This implies that the choice of bootstrapping method at the terminal state has relatively little impact: a policy is considered effective if it can generate a trajectory that reaches the time limit without violating any constraints. This observation is supported by the results in Figure 1 (c)-(d), where all bootstrapping methods perform similarly on the Hopper task. In contrast to manipulation tasks, where the reward function typically provides a smooth measure of distance to the target state, reward functions in locomotion tasks often only evaluate the quality of the current state and give little guidance regarding the optimal state. For example, in the Hopper task from Gymnasium, rewards favor states with higher forward velocity and penalize falling, which is less informative than the structured rewards in manipulation tasks. Moreover, the reward function does not directly reveal the goal state, nor does it guarantee that such a state even exists.. Consequently, using the Reward scheme to bootstrap the final value does not necessarily guarantee superior performance compared to other approaches.

**Influence of task horizon.** The success of vanilla policy gradient in Gymnasium suggests that RL problems can often be solved using a simple algorithm. As discussed earlier, value estimation is relatively robust in most long-horizon locomotion tasks, allowing a policy that directly follows the learned value patterns to perform effectively. In contrast, for certain manipulation tasks, the time limit $T$ is much shorter, requiring the policy to complete the task *within* $T$ steps. This requires the policy to be *near optimal under finite horizon*, since the standard RL objective corresponds to an infinite-horizon problem, where achieving the task within $T$ steps is not explicitly guaranteed. Therefore, the policy needs to be more aggressively optimized than in locomotion.

## 6 SOLVING VARIOUS CONTROL PROBLEMS WITH VPG

In this section, we validate our theoretical insights with extensive experiments across diverse control tasks, illustrating that a well-formulated policy optimization problem, when combined with an appropriate bootstrapping scheme, can always be solved by VPG, thereby supporting the core finding that accurate value estimation is the critical factor in deep RL.

**Implementation setup.** We evaluate three continuous control domains, each representing a distinct class of problems: Gymanisum (Brockman et al., 2016) for long-horizon sparse rewards, Maniskill3 (Tao et al., 2025) for short-horizon dense rewards with target states and IsaacLab (Makoviychuk et al., 2021) for long-horizon dense rewards. Regarding the PPO baseline in Gymnasium, we adapt the CleanRL implementation[1] (Huang et al., 2022). For Maniskill3[2] and IsaacLab[3], we directly use the corresponding built-in PPO implementation with default well-tuned configurations. In VPG, we disable the GAE factor. Each task is run with 5 random seeds, and the full hyperparameter settings for both algorithms are provided in Appendix A.

VPG, compared to PPO, reduces a set of parameters controlling policy update, such as the clipping ratio, number of epochs, and KL threshold, to a single parameter: the number of policy update steps. Note that when we have negative samples ($\hat{A}_t < 0$) in the VPG objective 1, performing gradient descent multiple steps on the corresponding $\log \pi_\theta(a_t|s_t)$ can lead to numerical instability, as the log-probability would be driven toward $-\infty$. To address this issue, we instead optimize the following objective for VPG:

$$L^{PG}(\theta) = \hat{\mathbb{E}}_{(s_t,a_t)\sim\pi_\theta}\Big[\log \pi_\theta(a_t|s_t) \ \max(\hat{A}_t, 0)\Big]. \tag{10}$$

The property $\nabla_\theta \hat{\mathbb{E}}\big[\log \pi_\theta(a_t|s_t)\big] = 0$ holds when $a_t \sim \pi_\theta(\cdot|s_t)$. However, this property immediately breaks if the policy distribution $\pi_\theta$ shifts: in particular, the action variance $\sigma$ can collapse rapidly after multiple update steps. To mitigate this issue, we perform only a single gradient update on the variance parameter $\sigma$, while allowing multiple updates on the mean parameter $\mu(\cdot; \theta)$, which is typically parameterized by a neural network.

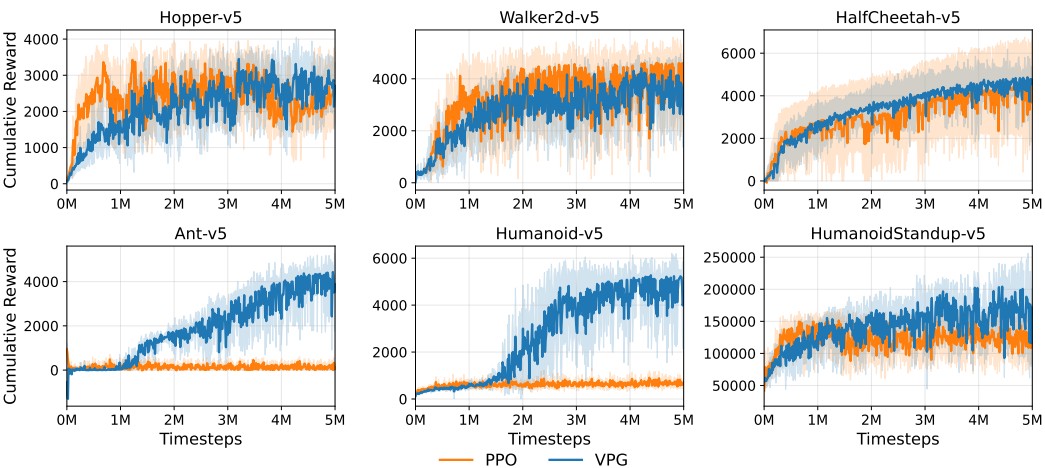

Figure 2: Performance of VPG and PPO on 6 *Gymnasium* tasks, measured by cumulative rewards.

**Gymnasium.** We first evaluate VPG and PPO on 6 sparse-reward locomotion tasks. For bootstrapping, we use the default None method in both algorithms. As shown in Figure 2, VPG matches the performance of PPO in Hopper, Walker2d, and HalfCheetah, and significantly outperforms it in higher-dimensional tasks such as Ant, Humanoid, and HumanoidStandup. This indicates that,

---

[1] https://github.com/vwxyzjn/cleanrl
[2] https://github.com/haosulab/ManiSkill
[3] https://github.com/leggedrobotics/rsl_rl

despite employing the same value estimation scheme, VPG is more stable and robust than PPO with respect to both algorithmic performance and dynamic complexity. Furthermore, while Gymnasium is a standard and canonical benchmark for deep RL, its control tasks are largely concentrated in one category: long-horizon, sparse-reward problems. As such, strong performance on Gymnasium should be interpreted with some caution, since it may not fully capture an algorithm's generality across broader classes of control tasks.

**Maniskill3.** Unlike Gymnasium tasks where the maximum rollout horizon is 1000, the manipulation tasks in ManiSkill3 have much shorter rollout lengths, ranging from 50 to 100. Moreover, these tasks feature more informative reward functions, as they directly measure the distance between the current and target states (see Section 5). Accordingly, we use the Reward method for bootstrapping on timeout in VPG, while the PPO baseline adopts the default Critic method. As shown in Figure 3, VPG consistently matches or outperforms PPO across all 6 tasks. It is also worth noting that although VPG with Reward bootstrapping performs well across all six tasks, careful tuning of the discount factor can be important. As shown in Figure 7, performance improves substantially when switching to the baseline's recommended $\gamma$, particularly on the more challenging tasks UnitreeG1TransportBox and PickSingleYCB.

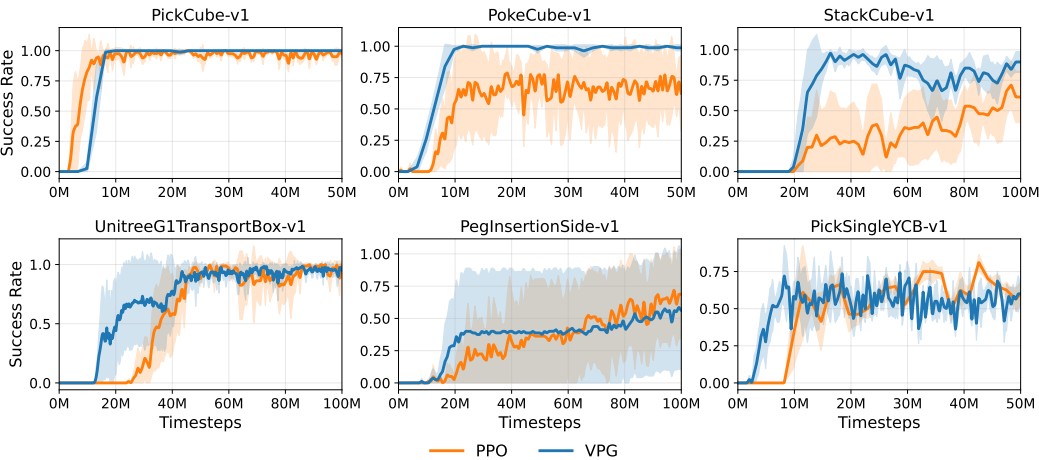

Figure 3: Performance of VPG and PPO on 6 *Maniskill3* tasks, measured by success rates.

**IsaacLab.** We now turn to control tasks from IsaacLab that differ from the previous two categories: they feature longer horizons than the manipulation tasks in ManiSkill3 and more complex rewards than the locomotion tasks in Gymnasium. The RSL-RL implementation employs the Critic bootstrapping scheme, which we also adopt for VPG in Repose-Cube-Shadow-Direct and Velocity-Rough-Anymal-C. For the Open-Drawer-Franka task, we apply the Reward bootstrapping method to VPG. Figure 1 presents the mean rewards of VPG and the well-tuned PPO baseline.

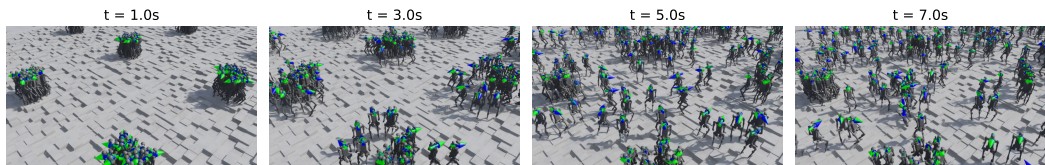

Figure 4: Snapshots of the Isaac-Velocity-Rough-H1 environment under the VPG policy with the None bootstrapping scheme, taken at $t = 1.0s, 3.0s, 5.0s,$ and $7.0s$.

We also perform a detailed comparison of VPG and PPO with all bootstrapping methods in Isaac-Velocity-Rough-H1. As shown in Figure 5, PPO's performance is sensitive to the choice of bootstrapping scheme, whereas VPG consistently converges to the same value across all three methods. This suggests that VPG is more robust than PPO with respect to value estimation. Interestingly,

despite being theoretically ill-posed, PPO with Critic outperforms the other variants. A possible explanation lies in the reward shaping: the task requires controlling the H1 robot to track a velocity field on rough terrain. Unlike the manipulation tasks in ManiSkill3, there is no universal target state applicable to all situations; instead, the reward function only evaluates the quality of the current state without prescribing how the agent should act next. In other words, the reward landscape is not aligned with the value landscape. Moreover, reward shaping is more complex than in Gymnasium tasks. For example, the reward of the H1 robot includes multiple regularization terms that penalize undesired behaviors such as deviations of the arms, ankles and torso. Therefore, simply reaching the maximum horizon does not guarantee the highest return. Although not optimal, VPG with the well-posed None scheme achieves reasonably good performance, leading to satisfactory agent behavior as shown in Figure 4.

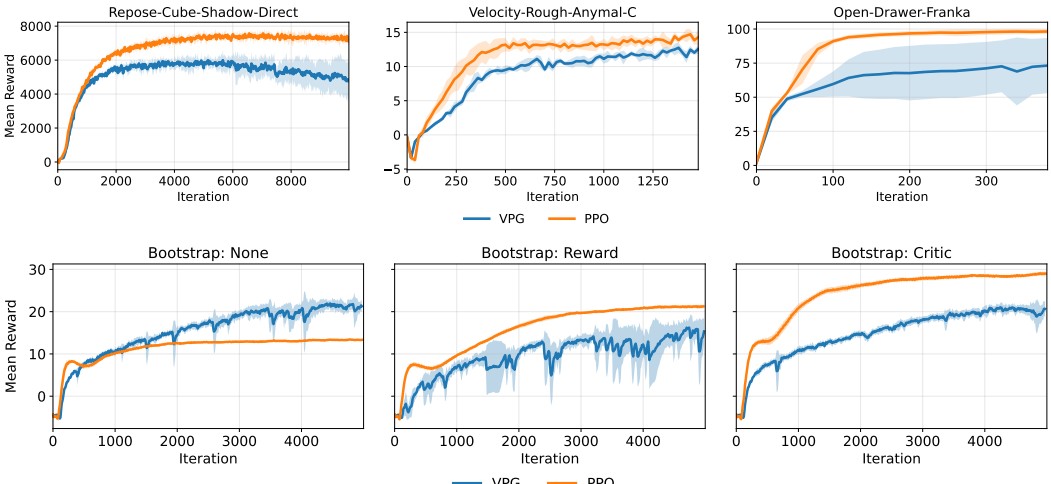

Figure 5: Top row: Mean reward over training steps for VPG and PPO on three robotic tasks in *IsaacLab*. Results are averaged over 5 random seeds, with solid lines denoting the mean and shaded regions representing $\pm 1$ standard deviation. Bottom row: Mean reward across training steps for VPG and PPO under different bootstrapping strategies (None, Reward, and Critic) in the Isaac-Velocity-Rough-H1 environment over 5 random seeds. Note that the mean reward is measured over the most recent 100 environment steps collected during training, instead of using the cumulative return of the deterministic policy as in prior benchmarks.

## 7 CONCLUDING REMARKS

In this work, we present an in-depth analysis of value estimation in policy gradient methods, with a particular focus on continuous control problems such as locomotion and manipulation. Our experiments demonstrate that, given an appropriate bootstrapping method, the generic VPG algorithm is sufficient to solve a wide range of challenging control tasks. These findings further substantiate and generalize the insight that accurate value estimation, rather than the enforcement of trust regions, is the key to the effectiveness of policy gradient methods. We emphasize, however, that this work is not intended to propose a new deep RL algorithm. Instead, our goal is to provide a systematic understanding of value estimation in policy gradient methods, which we believe can lead to more effective and principled approaches in the future.

**Limitations.** Although we provide an extensive study on how different classes of control problems can be effectively addressed with the corresponding bootstrapping schemes, our work does not explore the relationship between value estimation and reward shaping, which may also play a fundamental role in determining value accuracy. Regarding bootstrapping, we examined three methods in this paper, but it remains possible that alternative terminal reward specifications could perform even better. Finally, while our experiments span three distinct domains of control tasks, they do not cover the full spectrum of possible control problems, and there may exist tasks that fall outside the categories considered here.

REPRODUCIBILITY STATEMENT

The implementation details are provided in Section 6 and Appendix A. Task-specific configurations for IsaacLab are available in the official GitHub repository. For example, the configuration of Isaac-Velocity-Rough-H1 can be accessed at https://github.com/isaac-sim/IsaacLab/blob/main/source/isaaclab_tasks/isaaclab_tasks/manager_based/locomotion/velocity/config/h1/agents/rsl_rl_ppo_cfg.py . Similarly, configurations of ManiSkill3 tasks can be found at https://github.com/haosulab/ManiSkill/blob/main/examples/baselines/ppo/baselines.sh. All code will be released upon acceptance of this paper.

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

# A  EXPERIMENTAL SETTING

| Hyperparameter | Value |
| --- | --- |
| Discount factor ($\gamma$) | 0.99 |
| Batch size | 2048 |
| Optimizer | Adam, default |
| Horizon | 32 |
| Observation normalization | True |
| Actor learning rate | 0.001 |
| Critic learning rate | 0.001 |
| Actor network architecture | [64, 64] |
| Critic network architecture | [64, 64] |
| Num. value steps per iteration | 50 |
| Activation function | ELU |

Table 2: Hyperparameters for VPG in Gymnasium.

| Hyperparameter | Value |
| --- | --- |
| Discount factor ($\gamma$) | 0.99 |
| GAE factor ($\lambda$) | 0.95 |
| Batch size | 2048 |
| Minibatch size | 64 |
| Num. epochs | 10 |
| Optimizer | Adam, $\epsilon = 10^{-5}$ |
| Horizon | 32 |
| Clipping parameter | 0.2 |
| Entropy coefficient | 0 |
| Observation normalization | True |
| Advantage normalization | True |
| Reward normalization | True |
| Action clipping | True |
| Learning rate annealing | True |
| Clipped value loss | True |
| Actor learning rate | 0.0003 |
| Critic learning rate | 0.0003 |
| Actor network architecture | [64, 64] |
| Critic network architecture | [64, 64] |
| Activation function | ELU |
| Gradient clipping ($l_2$ norm) | 0.5 |

Table 3: Hyperparameters for PPO in Gymnasium.

| Hyperparameter | Gymnasium | Maniskill3 | IsaacLab |
|---|---|---|---|
| Value steps | 50 | 50 | 20 |
| Policy steps | 1 | 15 | 3 |
| Critic learning rate | 0.001 | 0.0001 | 0.001 |
| Actor learning rate | 0.001 | 0.0001 | 0.001 |
| Non-negative $\hat{A}_t$ only | No | Yes | Yes |

Table 4: Hyperparameters for VPG. Regarding other parameters such as network architecture and TD horizon, we use the same configuration as in PPO.

| Hyperparameter | Gymnasium | Maniskill3 | IsaacLab |
|---|---|---|---|
| Ratio clipping | Yes | Yes | Yes |
| Adaptive learning rate | No | No | Yes |
| Target KL | No | Yes | No |
| Entropy regularization | No | No | Yes |

Table 5: Hyperparameters for PPO. Other parameters such as learning rate and the number of epochs depend on their specific configuration.

**Adaptive learning rate in RSL RL.** While the default implementation in CleanRL does not use adaptive learning rate for PPO, it is employed in RSL RL that adaptively updates the learning rate every optimization step as follows:

```
if self.desired_kl is not None and self.schedule == "adaptive":
    with torch.inference_mode():
        kl = torch.sum(
            torch.log(sigma_batch / old_sigma_batch + 1.0e−5)
            + (torch.square(old_sigma_batch) + torch.square(old_mu_batch
− mu_batch)) / (2.0 ∗ torch.square(sigma_batch)) − 0.5, axis=−1,)
        kl_mean = torch.mean(kl)

    if kl_mean > self.desired_kl ∗ 2.0:
        self.learning_rate = max(1e−5, self.learning_rate / 1.5)
    elif kl_mean < self.desired_kl / 2.0 and kl_mean > 0.0:
        self.learning_rate = min(1e−2, self.learning_rate ∗ 1.5)
```

That is, the learning rate is multiplied by $1.5$ if the approximated KL divergence between the old and current policy is greater than twice of the desired KL divergence, and is divided by $1.5$ if the approximated KL divergence is smaller than half of the desired KL divergence.

**Target KL.** Maniskill3 sets a target KL divergence to further restrict the distance between the old and current policy. That is, PPO approximates the KL divergence every time the policy network is updated, and terminates the training on the current batch once the approximated KL exceeds the target KL.

# B ADDITIONAL EXPERIMENTAL RESULTS

In Section 6, we report the success rates of 6 ManiSkill3 tasks. Here, we present the corresponding cumulative returns in those experiments.

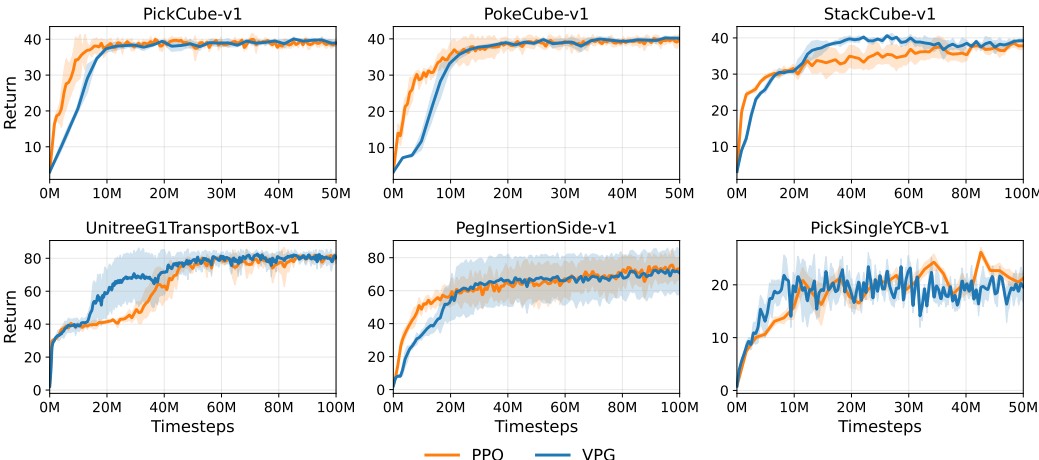

Figure 6: Corresponding cumulative returns of the Maniskill3 experiments in Figure 3.

The following plots illustrate the sensitivity of certain tasks to the choice of discount factor.

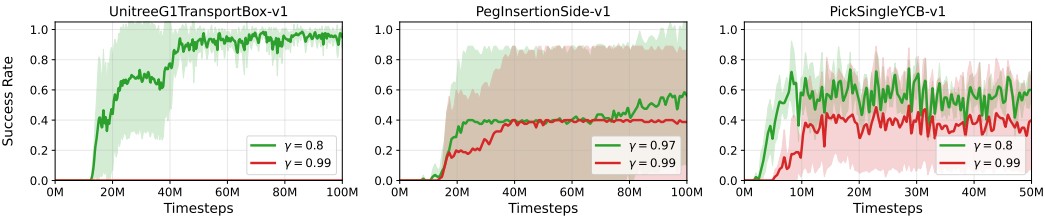

Figure 7: Success rate for the default $\gamma = 0.99$ and the recommended discount factors.

# C WALL-TIME EFFICIENCY

Aside from sample efficiency—which is commonly used as the primary efficiency metric in RL—the actual time required to solve a task is becoming increasingly relevant. With recent advances in simulation engines and GPU hardware, collecting large amounts of data can now take just a few minutes. As a result, sample efficiency may no longer be as critical as it once was. Instead, wall-clock time is gaining attention. As shown in Table 6, our algorithm requires significantly less simulation time than the default PPO implementation in CleanRL, while achieving comparable sample efficiency. This suggests that it is promising to explore methods that preserve performance while reducing the number of gradient steps in the training loop.

|     | Hopper | Walker2d | HalfCheetah | Ant  | Humanoid |
|-----|--------|----------|-------------|------|----------|
| VPG | 13.3   | 16.1     | 15.0        | 21.0 | 26.7     |
| PPO | 63.9   | 62.5     | 67.5        | 69.3 | 61.7     |

Table 6: The averaged wall-time (in minutes) consumed by each method in Gymnasium for 5M environment steps. The experiments are all run on a Dell G15 5520 laptop with an Intel Core i7-12700H CPU.

# D  PROOF OF THEOREM 4.1

Here we prove the general case of $H$-step TD estimation and the original theorem is immediate letting $H = 1$. Let $(\omega, \Sigma, P)$ denote the corresponding probability space. Let $\tau : \Omega \to \mathbb{N}$ be a stopping time, we have

$$\Omega = (\cup_{T=0}^{\infty}\{\tau = T\}) \cup \{T = \infty\}.$$

For any fixed $s \in \mathcal{S}$ and $\omega \in \Omega$, let

$$X(s; \omega) = \sum_{k=0}^{\tau(\omega) \wedge H - 1} \gamma^k R(s_k(\omega), a_k(\omega)) + \mathbf{1}_{\{\tau(\omega) \geq H\}} \gamma^{\tau(\omega) \wedge H} V_\phi(s_{\tau(\omega) \wedge H + 1}(\omega)) \qquad (11)$$

denote a sample path where $V_\phi$ is a solution of equation 6. Also, let

$$Y(s; \omega) = \sum_{k=0}^{\tau(\omega) - 1} \gamma^k R(s_k(\omega), a_k(\omega)) + \gamma^{\tau(\omega)} \Phi(s_\tau(\omega))$$

and it suffices to show that $X(s; \omega) = Y(s; \omega)$ a.s..

*Case 1:* When $\omega \in \{\tau \leq H\}$, we have

$$X(s; \omega) = \sum_{k=0}^{\tau(\omega) - 1} \gamma^k R(s_k(\omega), a_k(\omega)) + \gamma^{\tau(\omega)} \Phi(s_\tau(\omega)) = Y(s; \omega)$$

since the indicator function $\mathbf{1}_{\{\tau(\omega) \geq H\}}$ is zero.

*Case 2:* When $\omega \in \{\tau \geq H + 1\}$, there is a unique integer pair $(p, q)$ such that $\tau = pH + q$ with $p \in \mathbb{N}$ and $0 \leq q < H$. According to the recursion equation 11, it has

$$X(s; \omega) = \sum_{k=0}^{pH - 1} \gamma^k R(s_k(\omega), a_k(\omega)) + \gamma^{pH + 1} V_\phi(s_{pH}(\omega))$$

$$= \sum_{k=0}^{pH - 1} \gamma^k R(s_k(\omega), a_k(\omega)) + \gamma^{pH} \Big( \sum_{k=pH}^{\tau(\omega) - 1} \gamma^{k - pH} R(s_k(\omega), a_k(\omega)) + \gamma^{\tau(\omega) - pH} \Phi(s_\tau(\omega)) \Big)$$

$$= \sum_{k=0}^{\tau(\omega) - 1} \gamma^k R(s_k(\omega), a_k(\omega)) + \gamma^{\tau(\omega)} \Phi(s_\tau(\omega))$$

$$= Y(s; \omega).$$

Therefore, we have $X(s; \omega) = Y(s; \omega)$ for any $\omega \in \cup_{T=0}^{\infty}\{\tau = T\}$. Since $P(\tau < \infty) = 1$, it implies that $P(\{\tau = \infty\}) = 0$ and thus $X(s; \cdot) = Y(s; \cdot)$ a.s., which yields

$$V_\phi(s) = \mathbb{E}\Big[X(s; \cdot)\Big] = \mathbb{E}\Big[Y(s; \cdot)\Big] = \mathbb{E}_{a_t \sim \pi_\theta(\cdot | s_t)}\Big[ \sum_{k=0}^{\tau - 1} \gamma^k R(s_k, a_k) + \gamma^\tau \Phi(s_\tau) \Big]$$

and we complete the proof.

# LLM USAGE STATEMENT

We clarify that the use of LLMs in this paper is restricted exclusively to language polishing.

