# OpenReview forum: "Revisiting Value Estimation in Policy Gradient Methods"
_ICLR.cc/2026/Conference — Submitted to ICLR 2026_

### Official Review · Reviewer_hvCX · 2025-10-30

**Soundness:** 3
**Presentation:** 2
**Contribution:** 3
**Rating:** 6
**Confidence:** 3

**Summary:**

This paper notes that TD approximation is not guaranteed to converge to a unique solution when the ergodicity assumption is broken, and make the case that this is potentially a common problem in continuous spaces. The paper formalizes this notion in terms of well-posedness, that the solution of TD learning depends on how we assign value to episode terminations. The authors present two compelling examples of where this can be a problem, and a formal result that clarifies a sufficient condition for well-posedness. Empirical results demonstrate that a simple learning-free terminal value function can sometimes lead to better RL performance.

**Strengths:**

This paper presents a useful and, to the best of my knowledge, novel theoretical description of convergence in TD learning in terms of well-posedness. This work motivates its result well, describing the importance of accurate TD learning and a potential area of weakness. The formal result is strong and clear, and I have not seen it written in this language before. Also, I thought the two examples provided (the continuous one, and the simple description of how TD without termination can be under-determined) were very clever and clear. Overall, I think framing the TD learning problem this way is a valuable contribution to the field.

**Weaknesses:**

TD learning may be ill-posed when the values from terminal states are unconstrained or not defined, and the authors present two examples where this may be the case. However, in an episodic task with time-limits, states that are terminal for one episode may appear as part of a transition tuple in others. Therefore their values should be pinned by the bootstrapping update — and using a different “terminal reward” than value for those states could cause its own issues. This is, I would argue, the more common and realistic case. The absence of discussion about this feels like a big gap to me.

I’m not weighing this much because this is primarily a theory paper. But in the environments used for the empirical results, it’s likely that terminal states in one episode appear as intermediate states in another. And so it’s not clear that the modest improvements come from there, if the problems you tested on are also well-posed.

**Questions:**

Am I correct in understanding that TD is only ill-posed if some terminal states do not appear as non-terminal states elsewhere? I believe this paper would be much more complete with a discussion of this setting.

I believe that your result holds even in the case that s_T does also appear as a non-terminal state, but can you confirm this?

---

> ### Author Response · Authors · 2025-11-18
> **Rebuttal**
>
> Thank you for your positive feedback. For the question that
>
> > However, in an episodic task with time-limits, states that are terminal for one episode may appear as part of a transition tuple in others. Therefore their values should be pinned by the bootstrapping update — and using a different “terminal reward” than value for those states could cause its own issues.
>
> This is an excellent point. We agree that it is possible for a terminal state in one rollout trajectory to appear as an intermediate state in another trajectory. In such cases, the ill-posedness issue of the Critic method becomes less severe and may still allow it to produce good value estimates. However, even in this scenario, there will typically remain terminal states that do not appear in any other trajectories, and the value estimation at those states can still be ill-posed.
>
> Also, the severity of this issue heavily depends on the maximum episode length $T$. When the episode length is large, such as $T = 1000$ in IsaacLab, the situation you described is more likely to occur because each trajectory has a high probability of passing through states that serve as terminal states in other trajectories. In this case, the ill-posedness effect is outweighed by the bias introduced by the terminal value function, as shown in Figure 5.
>
> On the other hand, when the maximum episode length is small, such as $T = 50$ in Maniskill3, the trajectory has far fewer opportunities to cover a diverse set of states before reset. As shown in Figures 1 and 3, the ill-posedness issue of the Critic method becomes dominant in this regime, and using a terminal value function yields better performance.
>
> We hope that our response can address your concerns, and would be happy to discuss if you have any further questions.

---

> > ### Comment · Reviewer_hvCX · 2025-11-21
> > **Thank you for the reply**
> >
> > Thank you for the satisfactory reply. Some discussion of this would be helpful in the main paper I believe.

---

> > > ### Author Response · Authors · 2025-11-25
> > > **Thanks**
> > >
> > > We would also like to thank you once again for the constructive and insightful comments, which have provided valuable perspectives for improving the work.

---

### Official Review · Reviewer_q7FY · 2025-11-01

**Soundness:** 1
**Presentation:** 3
**Contribution:** 2
**Rating:** 2
**Confidence:** 4

**Summary:**

This paper investigates TD-based value estimation in policy gradient algorithms. They consider a case where there is no grounding for the value function (e.g., finite-length trajectories that don't reach a terminal state), observing that the Bellman equation has multiple solutions in such cases. They propose an alternate bootstrap target for truncated trajectories, show that it is well-posed, and validate the modification empirically.

**Strengths:**

* The well-posedness observation is really interesting, as it is relevant to cases where finite-length rollouts are collected from some starting state distribution. This setup is relatively common in deep RL, where an algorithm alternates between separate trajectory collection and batch-update phases. To my knowledge, it hasn't been widely acknowledged how this setup can readily break the standard assumptions made on the underlying MDP.

* The proposed modification does address the well-posedness concern in that it anchors the value estimates. The modification is somewhat heuristic, but the authors were p front about how the choice introduces an assumption on how the rewards must be consistent with the true values, and noted that this is (approximately) the case in the class of problems they considered.

**Weaknesses:**

* The demonstration of the three bootstrapping schemes was only done in one environment. Notably, the settings were changed from the defaults without discussion as to why. The novel "Reward" terminal modification was not investigated further, as section 6 only seems to compare PPO with the ill-posed target and VPG with the well-posed "None" scheme.

* There are considerable concerns around the empirical methodology. Only 5 seeds were used, which has been repeatedly shown to not be enough to make a proper statistical comparison for the claims being made (e.g., Henderson et al., 2017; Colas et al., 2018; Patterson et al, 2023; Patterson et al, 2024). Further, there is substantial variability in many of the plots, calling the statistical significance of the results into question—the shaded regions represent the standard deviation which is not a measure of confidence.

A minor concern that stood out but did not impact my score:

* The paper repeatedly suggests that the role of the value function in an actor-critic setup is not well understood. As worded, I'm not sure this is true, as hinted at by the paper citing evidence that the accuracy of the value estimates matter. In addition to being a baseline, values also help form an estimate of the return if TD is used, after which it is substituted where the return should be in the policy gradient theorem. Here, it is clear what role it is playing, and why it is beneficial for it to be accurate. Now, there is opportunity for handling settings when value-learing is ill-posed. That is where this paper makes a contribution, but statements suggesting that it's "unclear how the value estimation module, specifically the temporal-difference (TD) method (Sutton, 1988), operates within deep policy gradient methods" feel wrong without further specifics.

### References

* Henderson, P., Islam, R., Bachman, P. Pineau, J., Precup, D., Meger, D. (2018). Deep Reinforcement Learning that Matters.
* Colas, C., Sigaud, O., Oudeyer, P. (2018). How Many Random Seeds? Statistical Power Analysis in Deep Reinforcement Learning Experiments.
* Patterson, A., Neumann, S., White, M., White, A. (2023). Empirical Design in Reinforcement Learning.
* Patterson, A., Neumann, S., Kumaraswamy, R., White, M., White, A. (2024). The Cross-environment Hyperparameter Setting Benchmark for Reinforcement Learning

**Questions:**

* The well-posedness concerns seem orthogonal to their role within policy gradient methods. For example, the proposed modification is relevant to pure value-based methods as well, in setups where the value function similarly isn't anchored. I feel this makes the repeat emphasis on policy gradient methods somewhat imprecise. Could the authors clarify the choice to contextualize this in policy gradient methods?

* Can the authors elaborate the concern around time-independence? While the truncated return with no terminal reward is time-dependent, when it's applied to a non-finite-horizon problem, it is akin to optimizing over a moving window into the future from each state onward. While Figure 1 shows it performing poorly, it's unclear whether this is inherently problematic or a specific interplay with aggressive time limits. The remaining results with VPG, which I believe is using the "None" scheme, seems to be performing okay.

* Can the authors comment on the statistical significance of the results, and whether 5 seeds are enough for the conclusions drawn?

* If VPG uses a critic, advantage functions, and extra clipping for stability, is it even "vanilla" anymore? I initially assumed that VPG would be much closer to REINFORCE, as it is akin to direct application of the policy gradient theorem.

---

> ### Author Response · Authors · 2025-11-18
> **Rebuttal**
>
> Thank you for the detailed comments. Please see the following responses to your concerns.
>
> Weaknesses:
>
> > The demonstration of the three bootstrapping schemes was only done in one environment.
>
> Actually, the comparison of three bootstrapping schemes is not only conducted in one environment. We evaluate all three schemes in IsaacLab (Figure 5), Gymnasium, and ManiSkill (Figure 1), with each environment representing a class of control problems: Hopper for long-horizon simple-reward tasks, PickCube for short-horizon complex-reward tasks, and H1-Rough for long-horizon complex-reward tasks. We believe these examples are sufficient for illustration purposes, but we are happy to add more experiments to the draft.
>
> Regarding the default setting of PPO, we should clarify that it is not always ill-posed. For example, in Gymnasium the CleanRL implementation uses the None scheme for bootstrapping, which is well-posed according to our analysis.
>
> > Only 5 seeds were used, which has been repeatedly shown to not be enough to make a proper statistical comparison for the claims being made
>
> Regarding the concern on the number of random seeds, we agree that deep RL algorithms are sensitive and exhibit substantial variability. We chose to use 5 random seeds as it is consistent with common practice in the RL literature, including:
>
> * TRPO by Schulman et al.: 5 seeds per environment;
>
> * PPO by Schulman et al.: 3 seeds per environment;
>
> * SAC by Haarnoja et al.: 5 seeds per environment;
>
> * TD-MPC2 by Hansen et al.: 3 seeds per environment;
>
> * Even the paper “Implementation matters in deep policy gradients: A case study on PPO and TRPO” by Engstrom et al. uses 5 seeds per environment.
>
> We will increase the number of repeated runs in future revisions, but we still believe that using 5 random seeds is a reasonable choice.
>
> > The paper repeatedly suggests that the role of the value function in an actor-critic setup is not well understood. As worded, I'm not sure this is true, as hinted at by the paper citing evidence that the accuracy of the value estimates matter.
>
> We want to clarify that we only claim that the role of value estimation in policy gradient methods is not fully explored. Actor–critic methods include both on-policy and off-policy algorithms, and in Section 2 we explicitly state that “It has been clearly demonstrated that accurate value estimation is critical to the performance of off-policy algorithms…”. Our focus is specifically on on-policy methods. We will rephrase the statement to avoid any potential ambiguity.
>
> Questions:
>
> > The well-posedness concerns seem orthogonal to their role within policy gradient methods.
>
> Our analysis is focused on policy gradient (i.e., on-policy) methods because we aim to examine the particular way these methods approximate the value function—namely, through TD estimation. For off-policy methods such as TD3 and SAC, which approximate the value function (or Q function) directly from the Bellman equation, specifying the terminal value function is not as critical as in on-policy methods. These methods update the Q function using one-step transitions, which avoids the same ill-posedness issues present in policy gradient methods.
>
> > Can the authors elaborate the concern around time-independence?
>
> This is an important point. According to our analysis, time-independence is especially critical for short-horizon tasks, such as the manipulation tasks in ManiSkill. In the non-finite-horizon setting you mentioned, rollout trajectories never terminate due to a time limit, meaning there are no terminal states and all three bootstrapping schemes become equivalent. However, in practice, such settings are impossible due to real-world constraints, and our framework is designed precisely to address these practical issues.
>
> > Can the authors comment on the statistical significance of the results, and whether 5 seeds are enough for the conclusions drawn?
>
> Please refer to our response under Weaknesses. We adopted this setting from previous work.
>
> > If VPG uses a critic, advantage functions, and extra clipping for stability, is it even "vanilla" anymore?
>
> We adopt the definition of VPG from the OpenAI SpinningUp tutorial (https://spinningup.openai.com/en/latest/), which uses a critic and advantage functions. When referring to VPG, one usually means the original policy objective, in contrast to the surrogate objectives in TRPO and PPO. Regarding clipping, we want to clarify that unlike TRPO and PPO—where clipping is motivated by trust-region theory that limits the KL divergence between policies—our clipping is used solely to improve numerical stability and is independent of the RL framework itself.

---

> > ### Comment · Reviewer_q7FY · 2025-11-24
> > **Response to Rebuttal**
> >
> > > Actually, the comparison of three bootstrapping schemes is not only conducted in one environment. We evaluate all three schemes in IsaacLab (Figure 5), Gymnasium, and ManiSkill (Figure 1), with each environment representing a class of control problems: Hopper for long-horizon simple-reward tasks, PickCube for short-horizon complex-reward tasks, and H1-Rough for long-horizon complex-reward tasks. We believe these examples are sufficient for illustration purposes, but we are happy to add more experiments to the draft.
> >
> > My apologies—I missed that the bottom row of Figure 5 also compared the three schemes. I think my concern here echoes that of Reviewer fk1V, where it seemed to motivate and propose the reward-bootstrapping scheme, and then focus a lot of the evaluation on VPG. Perhaps it's a concern with how the paper is organized and the story it is trying to tell, but it seems to contradict all the prior motivation that we should be using reward-bootstrapping (e.g., time-independence).
> >
> > > Regarding the concern on the number of random seeds, we agree that deep RL algorithms are sensitive and exhibit substantial variability. We chose to use 5 random seeds as it is consistent with common practice in the RL literature...
> >
> > While it's definitely true that many works in the literature have used 3-5 seeds, the many references I mentioned explain why this is a problem and is leading to false claims and reproducibility issues in the field. Prior work doing it does not justify whether it is reasonable to support a paper's claim, as the paper is still responsible for the rationale behind it. For example, if a claim is that this is the first time any method has been able to solve some task, then a few runs can establish that. But if the claim is instead to say that one method is better than another on average, then many more runs are needed to convincingly demonstrate this. In all cases, proper empirical/statistical methodology is necessary. 3-5 is rarely enough for metrics based on central tendency, and concerningly, standard deviations were plotted which represent variability and not confidence in the mean. I'll restate the following question, as the references to prior work does not answer it:
> >
> > Can the authors comment on the *statistical significance of the results*, and whether 5 seeds are *enough for the conclusions drawn*?
> >
> > > We want to clarify that we only claim that the role of value estimation in policy gradient methods is not fully explored. Actor–critic methods include both on-policy and off-policy algorithms, and in Section 2 we explicitly state that “It has been clearly demonstrated that accurate value estimation is critical to the performance of off-policy algorithms…”. Our focus is specifically on on-policy methods. We will rephrase the statement to avoid any potential ambiguity.
> >
> > Can the authors clarify why accuracy is more important for off-policy methods than on-policy ones? I feel these are still orthogonal concerns. Depending on how the off-policy corrections are made, they can be made to be equivalently biased, and can be made to have equivalent dependence on the accuracy of the value estimates. Perhaps it is less on- vs. off-policy and emphasis on very specific algorithmic choices (e.g., one vs. multi-step TD, trajectory batches, transition-level replay buffers, etc.)
> >
> > > Our analysis is focused on policy gradient (i.e., on-policy) methods because we aim to examine the particular way these methods approximate the value function—namely, through TD estimation. For off-policy methods such as TD3 and SAC, which approximate the value function (or Q function) directly from the Bellman equation, specifying the terminal value function is not as critical as in on-policy methods. These methods update the Q function using one-step transitions, which avoids the same ill-posedness issues present in policy gradient methods.
> >
> > I still think terms are being needlessly conflated here as if they imply or constrain each other, when they are sound, stand-alone ideas. Policy gradient does not imply on-policy—there are a variety of off-policy policy gradient algorithms in the literature. TD estimation is also independent from policy gradient methods, as TD was original developed as a general policy evaluation method and can standalone as a value-based (i.e., non-policy gradient) algorithm. Because policy gradients can leverage value functions to reduce variance in the return estimates, TD can be used but is not required by definition in policy gradient or on-policy methods.

---

> > > ### Author Response · Authors · 2025-11-25
> > > **Reply**
> > >
> > > We thank the reviewer for the detailed response.
> > >
> > > > Perhaps it's a concern with how the paper is organized and the story it is trying to tell, but it seems to contradict all the prior motivation that we should be using reward-bootstrapping (e.g., time-independence).
> > >
> > > We agree that the narrative can be further refined. However, the main focus of the paper is clear: we revisit the value estimation module in policy gradient methods—examining when it is accurate and well-posed—and we show how properly constructed value estimates alone can provide strong algorithmic performance without any “modern approaches” such as trust-region heuristics.
> > >
> > > Regarding the reward bootstrapping method, we do not claim that it should always be used. Instead, we clearly demonsrate its strengths and limitations, including when it is appropriate and the rationale behind.
> > >
> > > Regarding the question on the number of random seeds, you mentioned that
> > >
> > > > For example, if a claim is that this is the first time any method has been able to solve some task, then a few runs can establish that. But if the claim is instead to say that one method is better than another on average, then many more runs are needed to convincingly demonstrate this.
> > >
> > > We would also like to note that nearly all of the papers cited in the rebuttal—along with many works in the RL literature—fall into the category of comparing methods and asserting improvements (e.g., TRPO and PPO over VPG, SAC over DDPG, and TDMPC2 over TDMPC and SAC). Our implementation setup follows this practice.
> > >
> > > > Can the authors comment on the statistical significance of the results, and whether 5 seeds are enough for the conclusions drawn?
> > >
> > > Unfortunately, we cannot definitively say whether five seeds are sufficient, and we believe most RL practitioners would reach the same conclusion. Moreover, we suspect that no fixed number of seeds can guarantee reproducibility in all cases. In this sense, the same question could be raised for any RL paper with empirical results, regardless of how many random seeds are used.
> > >
> > > > Can the authors clarify why accuracy is more important for off-policy methods than on-policy ones?
> > >
> > > We do not intend to suggest that value estimation accuracy is more important for off-policy methods than for on-policy methods. Our claims are simply that "It has been clearly demonstrated that accurate value estimation is critical to the performance of off-policy algorithms" and "the role of value estimation in policy gradient methods is not fully explored". We do not make, nor attempt to make, any comparison regarding the relative importance of value estimation between on-policy and off-policy methods.
> > >
> > > > Because policy gradients can leverage value functions to reduce variance in the return estimates, TD can be used but is not required by definition in policy gradient or on-policy methods.
> > >
> > > We agree that TD methods and policy gradient methods are independent, and this is precisely why we emphasize the context. Our point is that the TD estimation used in policy gradient methods is not the same as the TD schemes employed in approaches such as Q-learning or other off-policy algorithms.

---

### Official Review · Reviewer_e9Fn · 2025-11-01

**Soundness:** 3
**Presentation:** 3
**Contribution:** 3
**Rating:** 4
**Confidence:** 2

**Summary:**

This paper presents a re-evaluation of policy gradient methods, arguing that the accuracy and stability of value function (critic) estimation are far more crucial to performance than the specific policy (actor) update mechanisms, such as the clipping in PPO. The paper provides a theoretical framework for “well-posedness”, and its analysis and empirical validation. Empirically, the authors show that a modified Vanilla Policy Gradient algorithm, when paired with the theoretically-appropriate bootstrapping method for a given domain, can match or even outperform highly-tuned PPO baselines on challenging benchmarks.

**Strengths:**

- This paper challenges a core assumption in modern deep RL. The community has largely focused on the actor as the key to stability. This work provides a compelling argument that the critic and its proper formulation are the real bottleneck.
- The "well-posedness" framework, particularly the characterization of the "Critic" bootstrap as an under-determined system (Section 4), is insightful and provides a satisfying explanation for instabilities that many practitioners have likely observed.
- The experimental design is good and results are positive.

**Weaknesses:**

- The proposed reward bootstrapping method, while shown to be effective on ManiSkill3 tasks, assumes the reward landscape approximately reflects the ground-truth value function. This assumption holds for the selected manipulation tasks, which have well-shaped, dense rewards. Many important domains feature sparse rewards.
- The VPG implementation (Equation 10) is not quite vanilla. This "clipping" of negative advantages is a major deviation from true VPG. This modification is spiritually very similar to PPO's clipping, as it prevents the policy from being updated based on "bad" actions, which is known to stabilize training.

**Questions:**

-  The Reward bootstrapping method assumes the reward landscape is a good proxy for the value landscape. Could you elaborate on the specific failure modes of this method?
- What challenges would you have when applying the well-posedness framework and the proposed bootstrapping schemes to tasks with large, discrete action spaces?
- The paper builds on [1], which suggests more value updates are key. Your VPG uses 50 value steps. How does the number of value updates interact with well-posedness? Would an ill-posed method converge to the correct solution if you simply ran 1000 value updates instead of 50?

[1] Tao Wang, Ruipeng Zhang, and Sicun Gao. Improving value estimation critically enhances vanilla policy gradient. In ICML, 2025.

---

> ### Author Response · Authors · 2025-11-18
> **Rebuttal**
>
> Thank you for the important feedback. We hope that our response can address your concerns.
>
> For the weaknesses:
>
> > The proposed reward bootstrapping method, while shown to be effective on ManiSkill3 tasks, assumes the reward landscape approximately reflects the ground-truth value function. This assumption holds for the selected manipulation tasks, which have well-shaped, dense rewards. Many important domains feature sparse rewards.
>
> We agree that the efficacy of the Reward scheme depends on the specific reward landscape of a given problem (e.g., manipulation tasks), as mentioned in Line 249–252. We also believe that solving manipulation problems is itself important. According to the analysis in Section 5, for long-horizon sparse-reward tasks, the difference between bootstrapping schemes is not significant as shown in Figure 1.
>
> > The VPG implementation (Equation 10) is not quite vanilla. This "clipping" of negative advantages is a major deviation from true VPG. This modification is spiritually very similar to PPO's clipping, as it prevents the policy from being updated based on "bad" actions, which is known to stabilize training.
>
> This is a great point. The clipping in PPO is motivated by trust region theory, which suggests that the distance between two policies should be kept small to guarantee policy improvement. In our VPG implementation, the only purpose of clipping is to avoid numerical instability, which is independent of the RL framework and therefore much simpler and more straightforward. It also avoids the tedious hyperparameter tuning required in PPO, as discussed in Appendix A.
>
> For the questions:
>
> > The Reward bootstrapping method assumes the reward landscape is a good proxy for the value landscape. Could you elaborate on the specific failure modes of this method?
>
> It depends on the maximum episode length $T$: when the maximum episode length is large, such as $T = 1000$ in IsaacLab, the Critic method performs better since each trajectory has enough chance to go over states that may appear as terminal states in other trajectories. In this case, the effect of ill-posedness is outweighed by the bias of introducing a terminal value function, as shown in Figure 5.
>
> On the other hand, when the maximum episode length is small, such as $T = 50$ in Maniskill3, it significantly reduces the chance of covering sufficiently many states before reset. As in Figure 1 and 3, the ill-posedness issue of the Critic method becomes dominant and using a terminal value function performs better.
>
> > What challenges would you have when applying the well-posedness framework and the proposed bootstrapping schemes to tasks with large, discrete action spaces?
>
> Our observation is that the factors affecting the efficacy of each bootstrapping scheme is the reward landscape and the maximum episode length.
>
> > How does the number of value updates interact with well-posedness? Would an ill-posed method converge to the correct solution if you simply ran 1000 value updates instead of 50?
>
> Actually, the well-posedness affects the target value estimated by TD. That is, an ill-posed method may not converge to the correct solution regardless of the number of value steps. Having more value updates is indeed the key, given that the target values are well-posed and accurate, as this is sufficient to produce good value estimates on its own, which is exactly the case for the experiments in [1].

---

### Official Review · Reviewer_fk1V · 2025-11-03

**Soundness:** 2
**Presentation:** 3
**Contribution:** 2
**Rating:** 4
**Confidence:** 4

**Summary:**

This paper re-examines fundamentals of value estimation in continuous state space MDPs. It identifies a potential issue of the standard approach of using critics for value estimation and proposes an alternative for bootstrapping. Experiments in standard continuous control benchmarks show the benefits of the approach when combined with vanilla policy gradients compared to PPO.

**Strengths:**

- I appreciate the direction of the paper: Revisiting the fundamentals of the learning algorithms we use, in this case, value estimation in policy gradient methods. Better understanding the foundations can lead to more profound insights.

- In general, theory and experiments are clearly explained. The experiments are reasonably chosen to demonstrate the claims.

**Weaknesses:**

- I have some concerns about example 4.1 and the discussion of uniqueness of the value function.
The paper mentions that in continuous state spaces, the Bellman equation may not have a unique solution.
As far as I know, the uniqueness of solutions to the Bellman equation comes from the Banach fixed point theorem for contraction mappings. This theorem only requires that the (function) space under considerations is a complete metric space. These conditions are satisfied by the set of bounded, measurable functions equipped with the supremum norm. Thus, applying the Banach fixed point theorem would tell us that iteratively applying the Bellman equation would give us the unique value function in the limit.

This is consistent with example 4.1 because the alternative solution given is unbounded, which is not in the set of bounded, measurable functions. So, if we start with a bounded value function estimate, we will stay within the bounded set when applying the Bellman equation and recover the unique solution, the zero function. That is, we should not be encountering any issues.

Overall, I am unconvinced of the ill-posedness of the value estimation problem.


- Experimental sections: Some experiments are only run for particular environments but it would strengthen the claims if they were run across all the ones considered. For example, evaluating the bootstrapping strategy (None, Reward or Critic) was done in Fig.5 for the IsaacLab experiments but it would be more convincing if the three were run for the other environments. The reward bootstrapping seemed like a major part of the paper. These experiments could be added to the appendix at least.

- A minor point: I feel like the organization of the paper could be improved and there seems to be a slight lack of focus. There seems to be two main threads. One about the ill-posedness of value estimation with critics and an alternative using reward bootstrapping. A second about using vanilla policy gradient as an alternative to PPO. These felt slightly disjointed at times and perhaps they could be woven together more smoothly in the paper. Alternatively, focusing on one or the other could be good.

**Questions:**

- Line 228: There ia discussion of time-outs in RL environments and how bootstrapping is handled. An alternative that is notably absent is augmenting the state space with the time variable for finite-time MDPs. This can be a better alternative when the time index is important to the policy [1].


- Line 252: It is claimed that the proposed reward bootstrapping reflects the ground-truth $V^\pi$. Could this be confirmed experimentally?
- Also, it would be interesting to have a theoretical analysis. Are there conditions we could put on the reward and dynamics so the proposed method would be guaranteed to be effective?
The opposite is clear; there are simple, finite MDPs where reward boostrapping would fail to produce the correct optimal policy.


- Line 341: Clipping to eliminate updates on negative advantages was introduced. This seems like a nontrivial addition. If this is not a commonly-used technique, some more investigation would be welcome. For example, how does ths fare compared to entropy regularization, which is common addition preventing $\log \pi$ from going to $-\infty$.

Summary of review: \
This paper has some interesting ideas and explores some directions that I would be excited to see developed further. I am particularly interested in further investigations into value functions in continuous state spaces or alternatives to critic boostrapping. Currently, I am unconvinced of some of the main claims concerning ill-posedness of the value function and I think the experiments could be expanded to more strongly support the claims.
I would lean towards rejection at present.




Suggestions and other thoughts (no impact on score):
- It seems like there could be other interesting phenomena related to value functions in continuous state spaces. For example, [2] shows that a quite simple MDP with continuous states can lead to very complex value functions.
Perhaps it would be interesting to see if this complexity can lead to pitfalls in policy optimization?

- This paper [3] also explored how value estimates can impact the policy optimization path and may be interesting to consider. For example, perhaps choosing R(s)/(1-\gamma) as a terminal reward may be optimistic (greater than the true value function), which could be beneficial for exploration by increasing the policy's entropy.

- Is there a reason the estimator is called the vanilla policy gradient rather than REINFORCE (with baseline)?


[1] "Time Limits in Reinforcement Learning" Pardo et al.

[2] "The Gambler's Problem and Beyond" Wang et al.

[3] "Beyond variance reduction: Understanding the true impact of baselines on policy optimization" Chung et al.

---

> ### Author Response · Authors · 2025-11-18
> **Rebuttal**
>
> We thank the reviewer for their detailed feedback, and hope that our response can address the concerns raised.
>
> For the weaknesses:
>
> 1. As discussed in the Introduction (line 38–44), most existing theoretical guarantees on the uniqueness of the solution are established under the assumption that the state space is either finite or ergodic, which allows evaluation of the Bellman operator at every state. As you mentioned, the uniqueness theorem was proven for the set of bounded, measurable functions equipped with the supremum norm, which is impractical. The supremum norm of the distance between two functions $f_1$ and $f_2$ is the maximum difference $|f_1(s) - f_2(s)|$ over all possible $s$ in the state space. In other words, this norm cannot be accurately computed unless the difference is evaluated at all states, which is far from the case in most practical control problems.
>
>     Moreover, policy gradient methods do not approximate the value function in the same manner assumed in these theorems. Specifically, they estimate target values through rollout and bootstrapping rather than solving the Bellman equation via value iteration.
>
> 2. We compare three bootstrapping schemes in IsaacLab (Figure 5), Gymnasium, and ManiSkill (Figure 1) tasks, each representative of a distinct class of control problems: Hopper for long-horizon simple-reward tasks, PickCube for short-horizon complex-reward tasks, and H1-Rough for long-horizon complex-reward tasks. We believe these three examples suffice for illustration purposes but would be happy to add more experiments.
>
> 3. We agree that the narrative can be further refined. However, the main focus of the paper is clear: we revisit the value estimation module in policy gradient methods—examining when it is accurate and well-posed—and we show how properly constructed value estimates alone can provide strong algorithmic performance without any “modern approaches” such as trust-region heuristics.
>
> For the questions:
>
> * We explicitly discuss this method in Related Work, line 94–100.
>
> * The original claim is that the Reward bootstrapping scheme “is well-posed and time-independent, but relies on the assumption that the reward landscape approximately reflects the ground-truth value function.” This is an assumption about the structure of the reward landscape for a given control problem, not for the method itself.
>
> * As mentioned earlier, the proposed Reward method is expected to work when the reward landscape approximately reflects the ground-truth value function, which can be proven in the case of LQR problems. Although this theorem is feasible to include, we decided not to add it because the strong assumptions required would weaken its practical relevance.
>
> * This is a good point. The clipping in PPO is motivated by trust region theory, which suggests that the distance between two policies should be controlled to guarantee policy improvement. In our VPG implementation, the only goal of clipping is to avoid numerical instability, which is independent of the RL framework and much simpler and more straightforward. It also avoids the tedious hyperparameter tuning mentioned in Appendix A.
>
> For the suggestions:
>
> > This paper [3] also explored how value estimates can impact the policy optimization path and may be interesting to consider. For example, perhaps choosing R(s)/(1-\gamma) as a terminal reward may be optimistic (greater than the true value function)
>
> The idea of choosing $R(s)/(1 - \gamma)$ as the terminal reward function you suggested is exactly the Reward bootstrapping scheme we propose in the paper.
>
> > Is there a reason the estimator is called the vanilla policy gradient rather than REINFORCE (with baseline)?
>
> We follow the description of VPG from the OpenAI SpinningUp tutorial (https://spinningup.openai.com/en/latest/).

---

> ### Comment · Reviewer_fk1V · 2025-11-19
> **Response to Rebuttal**
>
> Thanks for the response.
> I would like to further discuss the theoretical aspects since those are a key component of the paper.
>
> >As you mentioned, the uniqueness theorem was proven for the set of bounded, measurable functions equipped with the supremum norm, which is impractical. The supremum norm of the distance between two functions $f_1$
>  and $f_2$
>  is the maximum difference $|f_1(s) - f_2(s)|$
>  over all possible $s$
>  in the state space. In other words, this norm cannot be accurately computed unless the difference is evaluated at all states, which is far from the case in most practical control problems.
>
> We do not have to practically evaluate the distance (supremum norm) between any two functions, this is only a theoretical tool used to prove the result. The uniqueness on the value function holds regardless of the practicality of computing the quantities.
>
> > Moreover, policy gradient methods do not approximate the value function in the same manner assumed in these theorems. Specifically, they estimate target values through rollout and bootstrapping rather than solving the Bellman equation via value iteration.
>
> This is a confusing point, since I thought the main concern brought up in the paper was about the soundness of the Bellman equation and uniquness of its solutions. The method used to estimate the value function would be a separate issue.
> In any case, policy gradient methods like VPG and PPO still usually learn a value function through TD updates, which are generally treated as a stochastic approximation to true value iteration (or its policy evaluation counterpart which uses a fixed policy).
>
> As such, while there may be other issues with estimating value functions in policy gradient methods, I don't think the argument concerning "non-uniqueness of the solutions to the Bellman equation" is valid.
>
> As a reference, this theorem shows that the solution to the Bellman equation is unique in the discrete-time, continuous state and bounded reward case [1] (see Proposition 4.2 p.55) .
>
> [1] ["Stochastic optimal control: the discrete-time case" Bertsekas and Shreve](https://books.google.ca/books?hl=en&lr=&id=1zM4EAAAQBAJ&oi=fnd&pg=PR3&dq=Dimitri+Bertsekas+and+Steven+E+Shreve.+Stochastic+optimal+control:+the+discrete-time+case,+volume+5.+Athena+Scientific,+1996.+&ots=SuOxlqyaXE&sig=dd1ydfBd_IKjjSfrW-kSBbCp_Vs&redir_esc=y#v=onepage&q=Dimitri%20Bertsekas%20and%20Steven%20E%20Shreve.%20Stochastic%20optimal%20control%3A%20the%20discrete-time%20case%2C%20volume%205.%20Athena%20Scientific%2C%201996.&f=false)
>
> Followup on other points: \
> For the questions:
> - Thanks for pointing this out, I had missed this reference to this method. In this case, I would still suggest trying adding the time variable as an input since it's a natural alternative. Perhaps it would have to be passed to the network similarly to sinusoidal position embeddings in transformers since neural networks are unlikely to function well with a single unbounded time variable as input.
> - For this point, I was referring to the specific claim that the reward bootstrapping method would work well because the reward matches the value function's structure. In the experiments, it is shown that the agent learns better with this proposed method. I am wondering if the original claim is in fact true (matching reward and value structures) or if there is some other reason for the observed benefits.
>
> For the suggestions:
> > The idea of choosing $R(s)/(1 - \gamma)$ as the terminal reward function you suggested is exactly the Reward bootstrapping scheme we propose in the paper.
>
> Indeed, I was not suggesting a new terminal reward, only offering a potential idea why it may be effective in practice.

---

> > ### Author Response · Authors · 2025-11-19
> > **Reply**
> >
> > We are happy to answer your questions.
> >
> > Regarding your concern about the theoretical results:
> >
> > > We do not have to practically evaluate the distance (supremum norm) between any two functions, this is only a theoretical tool used to prove the result. The uniqueness on the value function holds regardless of the practicality of computing the quantities.
> >
> > We agree that the assumptions used in theoretical analysis are not necessarily fully aligned with practical applications. However, in real control problems—where the underlying assumption is clearly violated because rollout data covers only a small portion of the state space—the theorem’s final claim of solution uniqueness is no longer reliable. Our goal, therefore, is to re-establish well-posedness under assumptions that are both realizable and more relevant in practice.
> >
> > > I thought the main concern brought up in the paper was about the soundness of the Bellman equation and uniqueness of its solutions
> >
> > That is precisely the main issue we aim to address in this paper. In fact, the well-posedness of the Bellman equation and the process by which the value function is estimated are tightly coupled rather than independent. For example, policy gradient methods typically learn a value function through TD updates, so it is natural to ask under what conditions this TD scheme will reliably converge to the true and unique value function. Our goal is not to disprove existing theoretical results, but to offer an application-oriented perspective that clarifies the practical challenges one may encounter when estimating the value function. In our view, uniqueness should be established under assumptions that are realistically achievable in practice.
> >
> > > As a reference, this theorem shows that the solution to the Bellman equation is unique in the discrete-time, continuous state and bounded reward case [1] (see Proposition 4.2 p.55).
> >
> > This theorem is one of the representative results we referred to earlier, establishing uniqueness of the solution using the infinity norm. While its assumptions are theoretically well-founded, they are not practical, as such conditions cannot be satisfied when running deep RL algorithms in practice. In contrast, our assumptions are trajectory-based, reflecting how policy gradient methods actually operate and collect data. The central element in our uniqueness theorem is the terminal value function, which naturally motivates the reward bootstrapping scheme we propose.
> >
> > For other points:
> >
> > > In this case, I would still suggest trying adding the time variable as an input since it’s a natural
> > alternative
> >
> > As discussed in the introduction, introducing time as an input violates the Markov property of the dynamics, which is a cornerstone of most deep RL pipelines. While this may appear to be a viable option, pursuing it further would take us beyond the main scope and direction of the current paper.
> >
> > > For this point, I was referring to the specific claim that the reward bootstrapping method would work well because the reward matches the value function's structure. In the experiments, it is shown that the agent learns better with this proposed method. I am wondering if the original claim is in fact true (matching reward and value structures) or if there is some other reason for the observed benefits
> >
> > As mentioned in the rebuttal, the original claim indeed holds in the LQR setting, and we chose not to include its proof because doing so might detract from the practical relevance of the paper. In Figure 1, the comparison between different schemes isolates the bootstrapping method as the only varying factor, thereby ruling out the influence of other confounding elements.

---

> ### Comment · Reviewer_fk1V · 2025-11-26
> **Reply**
>
> Thanks for the additional clarifications.
> I think I may be starting to see where there is a disconnect in our understanding of the theoretical results.
>
> To clarify my position, I am defining the value function as the expected discounted sum of rewards conditioned on a starting state. This is a function from states to real numbers that only depends on the underlying MDP and is independent to any learning algorithm. This value function also satisfies the (policy evaluation) Bellman equation.
> If I am understanding correctly, the authors agree that prior literature has established the uniqueness of this value function.
>
> > However, in real control problems—where the underlying assumption is clearly violated because rollout data covers only a small portion of the state space
>
> From this sentence, it seems like the authors are referring to a different "value function" defined only on the data that has been collected. If this is the case, then this brings us to the realm of offline RL, where there are theoretical results about when applying policy/value iteration on the collected data will give us good performance on the underlying MDP.
>
> This would not match the theoretical setting considered in the paper under review though, which only discusses the (true) value function in the MDP. In particular,the (counter)-example and theorem given in the paper make no distinction between collected data and the MDP.
> As such, I am still unsure of which issues are being discussed in the paper.
>
> > introducing time as an input violates the Markov property of the dynamics
>
> Concatenating the time as an input would not violate the Markov property. The time variable is independent (in a probabilistic sense) of the base state vector and also evolves in a Markovian way. We can directly verify that the Markovian property is still satisfied for the new Markov chain with its state expanded with the time index.

---

### Meta-Review · Area_Chair_npdm · 2025-12-24

**Summary:**

This paper proposes a "well-posedness" based framework to characterizes the TD estimation in policy gradient methods. Empirically, the authors show that with proper problem formulation and bootstrapping, even vanilla policy gradient can reliably solve the tasks, leading to more robust and interpretable deep RL methods.

The reviewers have raised several criticism/concerns about the paper that are not fully addressed, including:

(1) fk1V: not convinced of the ill-posedness concept.
(2) q7FY: insufficient experimental results (not enough random seeds).
(3) q7FY: the "vanilla policy gradient" implement is not vanilla, it is very different from what is analyzed.

The AC read the paper and also have some additional concern: The theoretically solid $\Phi(s)$ function exists but it is unknown. This gives the authors a chance to craft this function. The AC is not convinced that the performance is fully coming from the "well-posedness" rather than some sort of reward-shaping. Further, though the authors use a stopping time in the discussion, they seem to simply use a truncation time $T$ as the stopping time, which makes it unnecessary to introduce such a complication.

The AC carefully read the discussion and responses, and the AC do not think the authors fully address their concerns. In addition, by reading the paper, the AC also has some further concern in that the theoretically solid $\Phi(s)$ function exists but is unknown. This gives the authors a chance to craft this function. The AC is not convinced that the performance is fully coming from the "well-posedness" rather than some sort of reward-shaping. Further, though the authors use a stopping time in the discussion, they seem to simply use a truncation time $T$ as the stopping time, which makes it unnecessary to introduce such a complication.

Overall, based on the reviewers' discussion and the AC's own reading. We decide to reject this paper.

**Reviewer Concerns:**

There are too many concerns, the AC will only summarize the major ones.

Addressed: (1) fk1V's concern about Example 4.1 (only partially addressed)
                   (2) e9Fn's concern about the applicability to sparse reward environment (only partially addressed)
                   (3) q7FY's concern on the demonstration of the three bootstrapping schemes was only done in one environment.

Not addressed: see summary.

**Reviewer Scores:**

For the reviewers giving score of 4, their concerns are only partially addressed. The AC think there will be no change in score.

For the reviewer giving score of 6, she/he does not have major concerns, and will also not change the score.

---

### Decision · Program_Chairs · 2026-01-26

Reject